# What can Mamba do for 3D Volumetric Medical Image Segmentation?

## Abstract

Mamba, with its State Space Model (SSM), offers a more computationally efficient solution than Transformers for long-range dependency modeling. However, there is still a debate about its effectiveness in *high-resolution 3D medical image* segmentation. In this study, we present a comprehensive investigation into Mamba's capabilities in 3D medical image segmentation by tackling three pivotal questions: Can Mamba replace Transformers? Can it elevate multi-scale representation learning? Is complex scanning necessary to unlock its full potential? We evaluate Mamba's performance across three large public benchmarks—AMOS, TotalSegmentator, and BraTS. Our findings reveal that UlikeMamba, a U-shape Mamba-based network, consistently surpasses UlikeTrans, a U-shape Transformer-based network, particularly when enhanced with custom-designed 3D depthwise convolutions, boosting accuracy and computational efficiency. Further, our proposed multi-scale Mamba block demonstrates superior performance in capturing both fine-grained details and global context, especially in complex segmentation tasks, surpassing Transformer-based counterparts. We also critically assess complex scanning strategies, finding that simpler methods often suffice, while our Tri-scan approach delivers notable advantages in the most challenging scenarios. By integrating these advancements, we introduce a new network for 3D medical image segmentation, positioning Mamba as a transformative force that outperforms leading models such as nnUNet, CoTr, and U-Mamba, offering competitive accuracy with superior computational efficiency. This study provides key insights into Mamba's unique advantages, paving the way for more efficient and accurate approaches to 3D medical imaging. All code used in the experiments will be made publicly available.

## 1 Introduction

Volumetric medical image segmentation, which involves extracting 3D regions like organs, lesions, and tissues from modalities such as CT and MRI scans, is crucial for clinical applications like lesion contouring, diagnosis, and surgical planning. These tasks require not only local feature extraction but also the ability to capture long-range dependencies across entire volumes, which is vital for understanding the relationships between distant anatomical structures. While convolutional neural networks (CNNs), particularly U-Net (Ronneberger et al., 2015), have been foundational in 3D medical image segmentation (Zhou et al., 2019; Isensee et al., 2021; Zhang et al., 2020; Ye et al., 2023; Yu et al., 2020; Valanarasu et al., 2021), their limited receptive fields and locality biases hinder their capacity to model global context effectively. Transformers (Vaswani et al., 2017) address this with dynamic self-attention mechanisms, but their computational demands are impractical for handling large-scale, high-resolution 3D data.

With the introduction of the State Space Model (SSM), Mamba (Gu & Dao, 2023) offers a promising alternative for modeling long-range dependencies in 3D medical image segmentation. Unlike Transformers, Mamba achieves higher inference throughput and scales linearly with sequence length, making it a more computationally efficient solution. This efficiency makes Mamba particularly well-suited for the demands of 3D medical imaging, where high-resolution volumetric data requires both precision and speed to process large-scale structures effectively. Inspired by Mamba's success, a burgeoning body of work has sought to leverage its advantages for vision tasks, pioneering efforts such as Vision Mamba (ViM) (Zhu et al., 2024) and VMamba (Liu et al., 2024b). These

models employ multi-scan strategies, replacing the vanilla Mamba's single-scan approach, to allow long-range dependencies to manifest in multiple directions, improving the model's ability to capture spatial relationships in complex image data. As a result, several studies have explored replacing Transformers with Mamba blocks in 3D medical image segmentation. Notably, works like U-Mamba (Ma et al., 2024), SegMamba (Xing et al., 2024) and SwinUMamba (Liu et al., 2024a) have successfully integrated Mamba blocks as plugin modules within CNN architectures, achieving promising performance across various biomedical segmentation datasets. However, these efforts primarily demonstrate Mamba's feasibility without fully exploring its broader potential or its benefits.

To address this gap, in this work, we use three challenging 3D medical image segmentation benchmarks (*i.e.*, AMOS (Ji et al., 2022), TotalSegmentator (Wasserthal et al., 2023), and BraTS (Baid et al., 2021)) to conduct an in-depth exploration of Mamba's impact on 3D medical image segmentation, providing valuable insights for future research. Our investigation focuses on three aspects:

**Mamba's ability to replace Transformers** We aim to evaluate whether Mamba networks can replace Transformer-based architectures for long-range dependency modeling in 3D medical image segmentation, focusing on segmentation accuracy and computational efficiency. To this end, we designed two models: a Mamba-based network (UlikeMamba) and a Transformer-based network (UlikeTrans), both following a U-shaped encoder-decoder structure. Notably, we replace the original 1D depthwise convolutions (DWConv) (Chollet, 2017) in Mamba with 3D DWConv to better preserve volumetric data's spatial coherence. Our results show that UlikeMamba outperforms UlikeTrans in both accuracy and efficiency, especially with the 3D Mamba layer, while also avoiding the Out of Memory (OOM) issues faced by UlikeTrans.

**Mamba's capacity to enhance multi-scale representation learning** This section delves deeper into Mamba's potential for long-term dependency modeling to enhance multi-scale representation learning, a critical factor in achieving accurate 3D medical image segmentation. Successful volumetric segmentation requires the ability to capture both fine-grained details (such as small lesions or subtle tissue changes) and broader anatomical structures (such as large organs like the liver, heart, or kidneys). We design and implement four distinct multi-scale modeling schemes, and our results show that Mamba-based models excel at capturing and integrating multi-scale features. These models consistently demonstrate superior performance, especially in complex tasks like TotalSegmentator, which involves segmenting 117 anatomical structures, proving Mamba to be a versatile and robust solution for challenging 3D medical image segmentation scenarios.

**Whether complex multi-way scanning strategies are necessary?** Mamba's parallelized selective scan operation, designed for one-dimensional data, faces challenges when adapted to visual tasks. Many works, like Vision Mamba (Zhu et al., 2024) and VMamba (Liu et al., 2024b), introduce multi-way scanning mechanisms to preserve spatial coherence in vision tasks. To determine whether these complex scanning strategies are necessary for 3D medical image segmentation, we evaluate existing methods—single-scan (forward) and dual-scan (forward+backward)—and introduce two new approaches: dual-scan (forward+random) and Tri-scan (left-right, up-down, front-back). Dual-scan (forward+backward) offers minimal improvement due to strong structural priors in medical data. While dual-scan (forward+random) may capture complex dependencies, it risks distorting these priors, compromising segmentation precision. Tri-scan delivers the best performance by preserving comprehensive spatial relationships but incurs higher computational costs. Simpler scanning methods often suffice, with Tri-scan proving advantageous in more complex scenarios.

Our contributions are three-fold:

1. Rather than simply designing a new network, we conduct a thorough analysis of Mamba's role in 3D medical image segmentation, tailored to the specific challenges of the task, using three large, authoritative public datasets. This analysis provides strong insights and a foundation for future research in this domain.
2. We not only validate the effectiveness of existing strategies but also propose task-specific approaches, such as introducing 3D DWConv before SSM, developing multi-scale Mamba, and designing Tri-scan for 3D data, to further explore and enhance Mamba's capabilities for volumetric medical image segmentation.
3. Using validated strategies, we construct a Mamba-based network that sets a new benchmark for 3D medical image segmentation, outperforming advanced models such as CNN-based nnUNet, Transformer-based CoTr, UNETR and SwinUNETR, as well as existing Mamba-based U-Mamba, offering competitive accuracy with higher computational efficiency.

## 2 RELATED WORK

Mamba (Gu & Dao, 2023), known for its ability to capture long-range dependencies with superior memory efficiency and computational speed compared to Transformers, has gained traction in medical image segmentation. In this domain, U-Net and its variants dominate, however, integrating Mamba with CNN architectures has sparked interest, leading to the development of both hybrid and pure Mamba-based models. These efforts aim to harness Mamba's strengths in modeling global dependencies while maintaining the local feature extraction capabilities essential for segmentation.

In hybrid models, Mamba blocks are often combined with CNN-based architectures to balance the strengths of both methods. SegMamba (Xing et al., 2024) uses a multi-orientated Mamba module in the encoder, paired with CNN decoders. P-Mamba (Ye & Chen, 2024) integrates ViM (Zhu et al., 2024) blocks with noise suppression and local feature extraction, while Prompt-Mamba (Xie et al., 2024) incorporates prompt-based segmentation with ViM blocks. T-Mamba (Hao et al., 2024) enhances ViM blocks with frequency-based features, and U-Mamba (Ma et al., 2024) combines Mamba and CNNs in both the encoder and decoder, offering improved global context comprehension. Additionally, H-vmunet (Wu et al., 2024a) uses high-order interactions, and UltraLight VM-UNet (Wu et al., 2024b) optimizes multi-scale fusion with ViM layers and attention mechanisms.

Pure Mamba-based models rely on Mamba blocks either in the encoder, combined with a CNN decoder, or throughout the entire architecture. Swin-UMamba (Liu et al., 2024a) and LMa-UNet (Wang et al., 2024a) replace CNN blocks in the encoder with Visual State-Space (VSS) and bidirectional ViM blocks, capturing contextual information and refining pixel- and patch-level features. LightM-UNet (Liao et al., 2024) incorporates Residual Vision Mamba layers in both the encoder and bottleneck for better long-range spatial modeling. In fully Mamba-based architectures, both the encoder and decoder rely entirely on Mamba blocks. VM-UNet (Ruan & Xiang, 2024) was the first model to adopt this approach, using VSS blocks throughout. Mamba-UNet (Wang et al., 2024b) also employs a fully Mamba-based structure with VMamba blocks in the bottleneck, while TM-UNet (Tang et al., 2024) introduces Triplet SSM modules to fuse spatial and channel features, enhancing overall feature extraction.

The developments demonstrate the versatility of Mamba in medical image segmentation, offering a range of solutions w.r.t. specific tasks. However, these initial works primarily validate the feasibility of Mamba in this domain, lacking a comprehensive analysis of its impact and potential advantages.

## 3 MATERIAL

We use three publicly available volumetric medical image segmentation datasets to comprehensively evaluate the performance. These datasets are widely recognized as benchmarks in the medical image analysis community, covering a broad range of anatomical regions and imaging conditions:

**AMOS dataset** (Ji et al., 2022) The AMOS dataset consists of 300 abdominal CT scans collected from multiple centers and vendors, encompassing various imaging modalities and phases. Each scan is annotated at the voxel level for 15 abdominal organs, presenting a challenging test-bed for segmentation algorithms. Its diversity in disease cases, patient demographics, and imaging conditions makes it ideal for studying model robustness in real-world scenarios. In our experiments, we used the official training and validation sets.

**TotalSegmentator (TotalSeg) dataset** (Wasserthal et al., 2023) This dataset includes 1,228 CT images with annotations for 117 anatomical structures. The scans were randomly selected from clinical routines, offering a highly representative dataset that reflects real-world clinical conditions. The dataset spans a wide range of pathologies, scanners, sequences, and institutions, making it particularly well-suited for evaluating the generalizability of segmentation models. We used the official training and test sets in our experiments.

**BraTS 2021 challenge dataset** (Baid et al., 2021) The BraTS 2021 dataset includes 1,251 subjects, each with four 3D MRI modalities: native (T1), post-contrast T1-weighted (T1Gd), T2-weighted (T2), and T2 Fluid-attenuated Inversion Recovery (T2-FLAIR). It is a widely used benchmark for evaluating brain tumor segmentation algorithms, specifically for delineating tumor sub-regions such as enhancing tumor, necrosis, and edema, offering voxel-wise ground truth annotations provided by expert physicians. We split the dataset into 80% for training and 20% for testing.

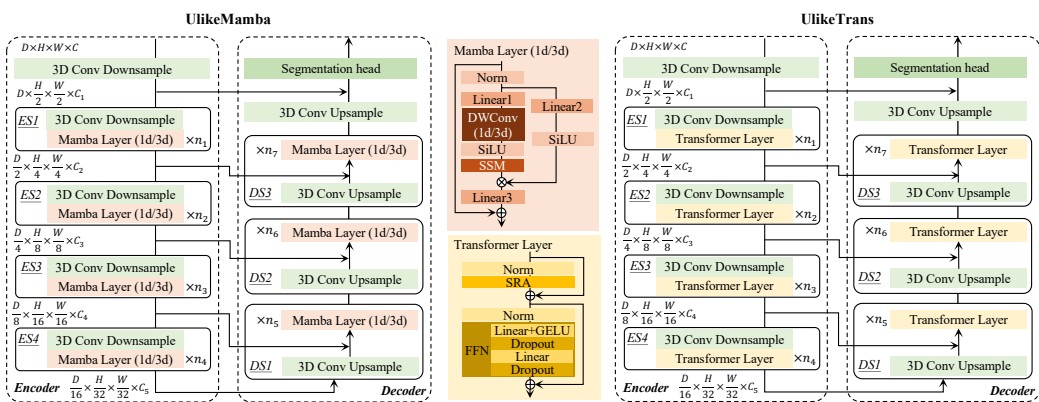

Figure 1: Mamba-based network (UlikeMamba) and Transformer-based network (UlikeTrans).

## 4 ANALYSIS 1: MAMBA VS TRANSFORMER

### 4.1 EXPERIMENTAL DESIGNS

This experiment is designed to compare the performance of U-shape Mamba- and Transformer-based networks, denoted as UlikeMamba and UlikeTrans, specifically for modeling long-range dependencies in volumetric medical image segmentation tasks. The goal is to evaluate both segmentation accuracy and computational efficiency, which is crucial for practical applications in clinical environments. As shown in Fig. 1, both UlikeMamba and UlikeTrans consist of an encoder $\mathcal{E}$ and a decoder $\mathcal{D}$. The corresponding blocks can be defined as

$$\begin{cases} \mathcal{E}_i := f_i \circ h_i & \mathcal{E}_i \in \mathcal{E} \\ \mathcal{D}_j := g_j \circ h_j & \mathcal{D}_j \in \mathcal{D}, \end{cases} \tag{1}$$

where $f_i$ and $h_i$ are the convolution layers and Mamba/Transformer layers, respectively, in the $i$-th block of the encoder, while $g_j$ and $h_j$ are the transposed convolution layers and Mamba/Transformer layers in the $j$-th block of the decoder. Concretely, in this section, we replace $h$ from Transformer to Mamba while keeping others unchanged.

**Mamba-based network** As illustrated on the left of Fig. 1, the Mamba-based network, referred to as UlikeMamba, adopts a U-shaped encoder-decoder architecture (Ronneberger et al., 2015). The encoder consists of a 3D convolutional (Conv) block and four stages (ES1 to ES4), with each stage composed of a 3D Conv block followed by a Mamba layer. This progressively downsamples the input through 3D Conv blocks, generating feature embeddings at each stage, which are then flattened and passed into the Mamba layers for sequential processing. These Mamba layers balance computational efficiency and feature extraction across multiple resolutions. The Mamba layer processes input through a series of operations. First, the input is normalized and passed through a linear layer for initial feature transformation. Depthwise convolutions (DWConv) are then applied to capture local spatial features, followed by a SiLU activation function to introduce non-linearity. The data is further processed by the state space model (SSM), which efficiently captures long-range dependencies with linear complexity. A residual connection merges the output from the SSM with earlier features, followed by further refinement via a final linear layer.

The decoder structure mirrors the encoder and consists of three stages (DS1 to DS3). Each stage upsamples the feature maps using 3D transposed Conv layers, followed by Mamba layers to refine the upsampled features. Skip connections link corresponding encoder and decoder stages to retain high-resolution, low-level information essential for accurate segmentation. The final segmentation head outputs the segmentation map through a 3D Conv upsampling layer. This overall architecture leverages the Mamba's strengths to efficiently process volumetric medical images while maintaining low computational overhead compared to more complex architectures. The specific architecture details can be found on the left of Figure 5 of Supplementary.

**Transformer-based network** The Transformer-based network, as shown on the right in Fig. 1, adopts a U-shaped encoder-decoder architecture similar to the Mamba-based network, but replaces

Table 1: Segmentation Dice scores (higher is better) and FLOPs (lower is better) of UlikeTrans and UlikeMamba across three test datasets. 'Parameters (Params)' and 'FLOPs' are calculated based on an input size of 128×128×128 and evaluated using an NVIDIA 3090 GPU.

| | AMOS | TotalSeg | BraTS | Average | Params (M) | FLOPs (G) |
|---|---|---|---|---|---|---|
| UlikeTrans_vanilla | OOM | OOM | OOM | OOM | 31.54 | OOM |
| UlikeTrans_SRA | 88.00 | 79.80 | 90.12 | 85.97 | 45.05 | 64.47 |
| UlikeMamba_1d (Vanilla) | 88.40 | 78.00 | 90.20 | 85.53 | 24.10 | 44.88 |
| UlikeMamba_3d | **89.45** | **82.60** | **90.29** | **87.45** | 24.30 | 46.03 |

the Mamba layers with Transformer layers, hence referred to as UlikeTrans. Each Transformer layer consists of a self-attention module and a feed-forward network (FFN) with two hidden layers. Initially, we experimented with vanilla point-to-point self-attention, however, this approach resulted in extreme computational complexity and excessive memory usage when applied to 3D volumetric images, making quantitative comparisons impractical. To address this, we implemented the spatial-reduction attention (SRA) layer (Wang et al., 2021) to reduce spatial complexity and enable UlikeTrans to handle high-resolution volumetric medical images for comparisons. Given a query $q$, a key $k$, and a value $v$ as the input, SRA first reduces the spatial resolution of $k$ and $v$, and then feeds $q$, reduced $k$, and reduces $v$ to a multi-head self-attention layer to produce refined features. The specific architecture details can be found on the right of Figure 5 of Supplementary.

## 4.2 Training setup and evaluation metrics

Both UlikeMamba and UlikeTrans were implemented using the nnUNet (Isensee et al., 2021) framework, which automatically selects batch sizes and patch sizes tailored to each dataset. We utilized the AdamW optimizer (Loshchilov & Hutter, 2018) with an initial learning rate of 0.0001. All networks were trained for 1000 epochs, with each epoch consisting of 250 iterations. To evaluate the segmentation results quantitatively, we calculated the Dice coefficient (Dice), a metric measuring the overlap between the predicted segmentation and the ground truth. Additionally, we computed the floating-point operations per second (FLOPs) to assess the computational complexity of each model. Ideally, higher Dice scores indicate better segmentation accuracy, while lower FLOPs reflect greater computational efficiency.

## 4.3 Results and Analysis

**Directly using vanilla Mamba** The results in Table 1 show that UlikeMamba_1d, using the vanilla Mamba layer with DWConv 1D, performs competitively across all datasets, achieving Dice scores similar to UlikeTrans_SRA, while requiring fewer parameters and computational resources (44.88 GFLOPs vs. 64.47 GFLOPs). UlikeMamba_1d avoids the Out of Memory (OOM) issues faced by the vanilla UlikeTrans model, which is hindered by the excessive memory demands of point-to-point self-attention for 3D volumetric data. This highlights the efficiency of Mamba in handling long-range dependencies while maintaining a low computational footprint, making it especially suited for resource-constrained environments.

The main reason is that Transformers are limited by memory capacity and complexity at higher resolutions and cannot be used directly. Moreover, when sequences are too long, establishing point-to-point relationships makes it difficult to effectively focus on key information. Mamba's sequence modeling combined with memory modules gives it certain advantages in volume segmentation, where longer sequence modeling is required. Besides, the ability of Mamba networks to achieve comparable or even superior Dice scores to Transformer models across the datasets (AMOS, TotalSeg, BraTS) indicates their proficiency in capturing long-range spatial relationships within the data. This is particularly significant given that medical image segmentation often relies on the precise delineation of complex anatomical structures that may be distributed sparsely across the image space. The Mamba model's performance suggests that its architecture can effectively encapsulate these relationships without the need for extensive computational resources.

**DWConv 1D vs. DWConv 3D** We noted that in vanilla Mamba layer (Gu & Dao, 2023) and Vision Mamba (Zhu et al., 2024), DWConv 1D with a kernel size of 4 is used. However, in Mamba, the

input feature embeddings are flattened and processed sequentially, causing DWConv 1D to disrupt the original 3D spatial structure. This sequential processing links distant voxels while neglecting immediate neighbors in the 3D space, undermining spatial coherence essential for accurate segmentation. To address this, we replace DWConv 1D with DWConv 3D in establishing 3D priors, ensuring local features are captured across all dimensions. This adjustment preserves the 3D structure of volumetric medical images, allowing the network to capture both local details and global context better. As shown in Table 1, Mamba 1D performs on par with Transformer (average Dice score: Transformer 85.97 vs. Mamba 1D 85.53), while the Mamba 3D improves result from 85.53 to 87.45 and consistently outperforms Mamba 1D across all the datasets with only a slight increase in parameters and FLOPs. This proves our above claims and further demonstrates that Mamba is not only effective but also has the potential to exceed the capabilities of Transformer in volumetric medical image segmentation tasks.

## 5 ANALYSIS 2: MAMBA'S POTENTIAL IN MULTI-SCALE MODELING

### 5.1 EXPERIMENTAL DESIGNS

In the first section, we establish that Mamba could effectively replace Transformers for long-range dependency modeling in volumetric medical segmentation tasks. This section aims to delve deeper into the potential of Mamba and investigate whether its long-term dependency modeling can significantly enhance multi-scale representation learning—a critical aspect of accurate volume segmentation. Multi-scale modeling plays a crucial role in medical image segmentation, where structures vary in size and capturing both fine details and broader anatomical context is essential.

While the pyramid structure captures features at different resolutions, we are further inspired by (Szegedy et al., 2015) to use multiple receptive fields within each resolution feature map to capture details at varying levels. Small receptive fields focus on fine structures like lesions, while larger receptive fields capture broader context, such as organ boundaries and anatomical regions. We design and implement four distinct multi-scale modeling schemes (Fig. 2). These schemes explore different strategies for fusing features from multiple receptive fields, leveraging Mamba's and Transformer's long-range dependency modeling capabilities for multi-scale representation learning. Specifically, we replace the whole blocks $\mathcal{E}_i$ and $\mathcal{D}_j$ in Eq. (1) with the following different multi-scale ones:

**MSv1** This model combines two parallel convolution layers with different kernel sizes ($3 \times 3 \times 3$ and $7 \times 7 \times 7$) to extract multi-scale features. These features are then processed through either parallel Mamba or Transformer layers. The outputs are integrated via element-wise summation, allowing efficient fusion of local and global information from different receptive fields.

**MSv2** In this configuration, the outputs from the $3 \times 3 \times 3$ and $7 \times 7 \times 7$ convolutions are concatenated before being processed by either Mamba or Transformer layers. This structure ensures that the multi-scale information is integrated at an earlier stage, allowing long-range dependency modeling to operate on a richer set of features.

**MSv3** This scheme extends the multi-scale feature extraction by incorporating an additional convolution layer with a $5 \times 5 \times 5$ kernel, alongside the $3 \times 3 \times 3$ and $7 \times 7 \times 7$ convolutions. The outputs are concatenated and then passed through Mamba or Transformer layers. The inclusion of the intermediate $5 \times 5 \times 5$ convolution provides an additional scale, improving the granularity of multi-scale feature extraction.

**MSv4** Designed specifically for the Mamba layer, MSv4 incorporates three DWConv 3D layers ($3 \times 3 \times 3$, $5 \times 5 \times 5$, and $7 \times 7 \times 7$) to extract multi-scale features. These are then concatenated and processed by Mamba's state space model (SSM), which captures long-range dependencies while maintaining 3D spatial integrity. MSv4 aims to maximize Mamba's ability to process both local and global features, taking full advantage of its sequence modeling capabilities to handle complex volumetric data efficiently.

To evaluate the effectiveness of these multi-scale modeling strategies, we systematically replace the encoder stages (ES1 to ES4) in both UlikeTrans_SRA and UlikeMamba_3d with the proposed MSv1, MSv2, and MSv3 schemes. MSv4, due to its specific design for Mamba, was applied only to the UlikeMamba_3d model. This design allows us to directly compare the performance of Mamba and

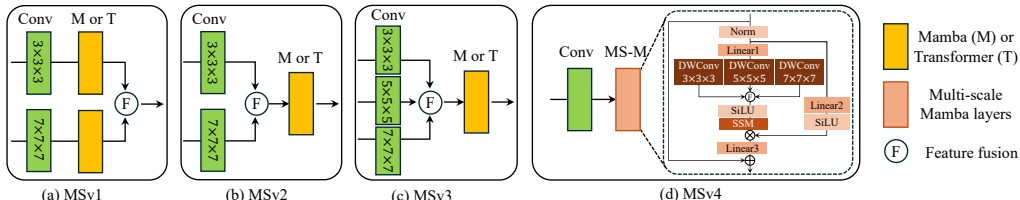

(a) MSv1    (b) MSv2    (c) MSv3    (d) MSv4

Figure 2: Four multi-scale modeling schemes for evaluating and comparing the long-range dependency modeling capabilities of Mamba and Transformers for multi-scale representation learning.

Table 2: Segmentation Dice scores (higher is better) and FLOPs (lower is better) of UlikeTrans and UlikeMamba with different multi-scale strategies across three test datasets.

|  | AMOS | TotalSeg | BraTS | Average | Params (M) | FLOPs (G) |
|---|---|---|---|---|---|---|
| UlikeTrans_SRA | 88.00 | 79.80 | 90.12 | 85.97 | 45.05 | 64.47 |
| UlikeTrans_SRA with MSv1 | 88.49 (+0.49) | 82.40 (+2.60) | 90.21 (+0.09) | 87.03 (+1.06) | 88.02 | 139.28 |
| UlikeTrans_SRA with MSv2 | 88.87 (+0.87) | 82.40 (+2.60) | 90.43 (+0.31) | 87.23 (+1.26) | 47.83 | 116.59 |
| UlikeTrans_SRA with MSv3 | 88.78 (+0.78) | 82.70 (+2.90) | 90.31 (+0.19) | 87.26 (+1.29) | 49.03 | 135.71 |
| UlikeMamba_3d | 89.45 | 82.60 | 90.29 | 87.45 | 24.30 | 46.03 |
| UlikeMamba_3d with MSv1 | 89.43 (-0.02) | 83.20 (+0.60) | 90.09 (-0.20) | 87.57 (+0.12) | 55.13 | 112.50 |
| UlikeMamba_3d with MSv2 | 89.33 (-0.12) | 83.40 (+0.80) | 90.52 (+0.23) | 87.75 (+0.30) | 27.09 | 98.16 |
| UlikeMamba_3d with MSv3 | 89.50 (+0.05) | 83.70 (+1.10) | 90.40 (+0.11) | 87.87 (+0.42) | 28.29 | 117.28 |
| UlikeMamba_3d with MSv4 | 89.48 (+0.03) | 84.50 (+1.90) | 90.06 (-0.23) | 88.01 (+0.56) | 31.57 | 62.23 |

Transformer layers in the context of multi-scale modeling. By testing Mamba and Transformer in MSv1, MSv2, and MSv3, we can determine which architecture better exploits multi-scale features for long-range dependency modeling. Since MSv4 is specifically designed to leverage Mamba's capabilities, it is used solely to evaluate Mamba's efficiency in handling complex 3D medical data.

## 5.2 RESULTS AND ANALYSIS

The results of both UlikeTrans_SRA and UlikeMamba_3d architectures, incorporating different multi-scale receptive field modeling schemes, are summarized in Table 2.

**Comparison of multi-scale schemes on UlikeTrans_SRA and UlikeMamba_3d** Both Ulike-Trans_SRA and UlikeMamba_3d show improvements with the application of multi-scale receptive field modeling, but UlikeMamba_3d consistently outperforms UlikeTrans_SRA in terms of segmentation accuracy and computational cost. For example, UlikeMamba_3d with MSv4 achieves the highest average Dice score of 88.01 while maintaining 62.23 GFLOPs, significantly better than the 116.59 GFLOPs required by UlikeTrans_SRA with MSv2 achieving the Dice score of 87.23.

Interestingly, the performance gains from multi-scale strategies are more noticeable in Ulike-Trans_SRA. For instance, UlikeTrans_SRA improves from 85.97 to 87.23 with MSv2, while Ulike-Mamba_3d shows a smaller improvement from 87.45 to 87.75. This may be because Ulike-Trans_SRA has lower initial performance, so it gains more from multi-scale modeling, which helps overcome self-attention's limitations in capturing long-range dependencies in high-resolution data. In contrast, UlikeMamba_3d is already efficient at modeling long-range dependencies through its SSM, which is well-suited for high-resolution volumetric data. As a result, Mamba-based models see relatively smaller gains from multi-scale strategies since they are already effective at capturing fine details and broader context through their long-term sequence modeling.

**Task-specific impact** The performance improvements for multi-scale schemes are most evident in the TotalSeg dataset for both UlikeTrans_SRA and UlikeMamba_3d. For instance, UlikeTrans_SRA improves from 79.80 (baseline) to 82.40 (MSv2), while UlikeMamba_3d improves from 82.60 to 84.50 with MSv4. This is in contrast to smaller gains observed on AMOS and BraTS. The TotalSeg dataset with a larger-scale data size requires the segmentation of 117 anatomical classes, making it much more complex than AMOS (with 15 organs) and BraTS (focused on three brain tumor sub-regions). The presence of a wide range of structures in TotalSeg—varying in size from fine tissues to

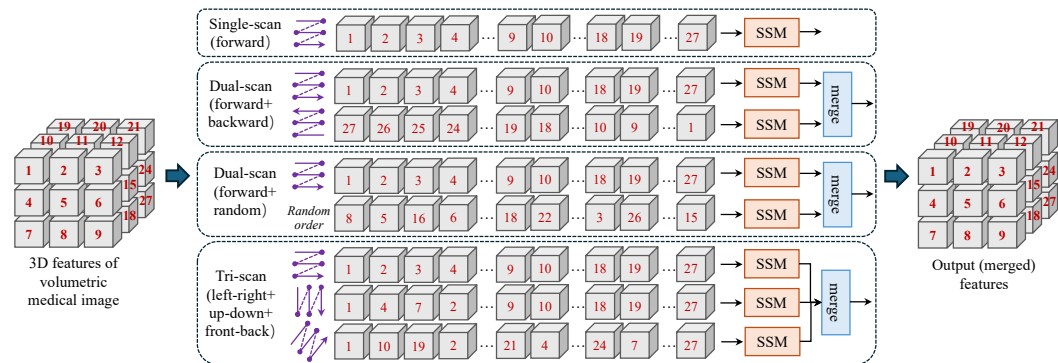

Figure 3: UlikeMamba_3d with different sequential scanning strategies.

large anatomical structures—makes multi-scale feature extraction particularly important. The ability to capture both fine-grained and large-scale structures is crucial, and this is where the integration of multi-scale receptive fields brings significant performance improvements. In contrast, AMOS and BraTS deal with fewer segmentation classes, where a single receptive field might suffice for most features, resulting in more modest performance gains.

**Strength of MSv4** Our proposed MSv4, specifically designed for Mamba, optimizes multi-scale feature extraction in 3D medical data. It delivers the best overall performance across datasets while maintaining lower computational costs. With MSv4, UlikeMamba_3d achieves the highest Dice score on TotalSeg (84.50) and remains competitive on AMOS and BraTS, all with fewer FLOPs than other multi-scale schemes. MSv4's design excels by fully leveraging Mamba's SSM for efficient long-range dependency modeling, integrating multi-scale features with minimal overhead, making it ideal for complex volumetric segmentation tasks.

## 6 ANALYSIS 3: MULTI-SCAN STRATEGY VS SINGLE-SCAN STRATEGY

### 6.1 EXPERIMENTAL DESIGNS

Mamba's core mechanism, particularly its parallelized selective scan operation, was originally designed for one-dimensional sequential data processing. This introduces potential challenges when adapting it to visual data, where spatial components are not inherently sequential. To address this, Vision Mamba (Zhu et al., 2024) and Vmamba (Liu et al., 2024b) propose multi-way scanning mechanisms tailored to preserve spatial coherence in vision tasks. The goal here is to assess whether these complex scanning strategies are needed or if simpler approaches suffice for volumetric medical image segmentation, where maintaining spatial relationships between voxels is critical for accuracy.

To investigate, we conducted experiments using the same backbone architecture UlikeMamba_3d but varied the scanning mechanism to evaluate its effect on segmentation performance. In other words, we consider different scanning strategies for UlikeMamba_3d by modifying it in $h$ of Eq. (1) only. We implemented the following scanning strategies, as shown in Fig. 3: **Single-scan** (Gu & Dao, 2023), proposed in vanilla Mamba, processes 3D features by flattening the volumetric features and scanning them sequentially along a single axis, typically in the forward direction. **Dual-scan (forward + backward)**, proposed in Vision Mamba (Zhu et al., 2024), processes 3D input by scanning twice along the same axis—once in the forward direction and once in the backward direction. The features from both scans are then merged, allowing the model to incorporate information from both directions along the chosen axis. This method maintains the same backbone structure but introduces bidirectional data flow in Mamba layers to capture more comprehensive spatial information. **Dual-scan (forward + random)** is a new approach that combines a standard forward scan with an additional scan in a random order. This method introduces variation in the scanning sequence to capture a broader range of spatial relationships, while still preserving the overall sequential structure. The features from forward and random scans are merged to create a more diverse feature representation of the volumetric input. **Tri-scan**, inspired by Vmamba (Liu et al., 2024b) and adapted for 3D medical volumetric data, scans the input in three directions: left-right, up-down, and front-back.

Table 3: Segmentation Dice scores (higher is better) and FLOPs (lower is better) of UlikeMamba with different sequential modeling scanning strategies across three test datasets.

| | AMOS | TotalSeg | BraTS | Average | Params (M) | FLOPs (G) |
|---|---|---|---|---|---|---|
| Single-scan | 89.45 | 82.60 | 90.29 | 87.45 | 24.30 | 46.03 |
| Dual-scan (forward + backward) | 89.74 (+0.29) | 83.00 (+0.40) | 90.27 (-0.02) | 87.67 (+0.22) | 25.34 | 49.56 |
| Dual-scan (forward + random) | 89.42 (-0.03) | 83.30 (+0.70) | 90.08 (-0.21) | 87.60 (+0.15) | 25.34 | 49.56 |
| Tri-scan | 89.77 (+0.32) | 83.60 (+1.00) | 90.43 (+0.14) | 87.93 (+0.48) | 26.38 | 53.09 |

Each scan generates a sequence of features along its respective axis. These features are then passed through separate SSM layers for further processing, and the outputs are merged to form a unified representation of the 3D volume.

## 6.2 RESULTS AND ANALYSIS

**Results** The experimental results are summarized in Table 3. Across the three datasets (AMOS, TotalSeg, and BraTS), we observe that Tri-scan achieves the highest average Dice score (87.93) but comes at the cost of increased computational complexity, as indicated by its higher parameter count (26.38M) and FLOPs (53.09G). The Dual-scan (forward + backward) approach performs slightly better than Single-scan and Dual-scan (forward + random), with an average Dice score of 87.67 and 87.60 respectively. However, the performance gains for dual-scan methods over single-scan are marginal. The Single-scan method, while having the lowest computational requirements (24.30M parameters, 46.03G FLOPs), still delivers competitive performance with an average Dice score of 87.45, closely trailing the more complex scanning mechanisms.

**Analysis** The Dual-scan (forward + backward) method aims to help the model capture spatial information from both the start and end of the sequence, potentially building a more complete data representation. However, in our task, the improvement over Single-scan is slight probably because 3D medical images have strong structural priors, allowing most key spatial relationships to be captured effectively by a unidirectional scan. The added backward scan introduces limited sequential diversity, failing to uncover significantly more data patterns. Besides, Mamba's long-range dependency modeling is already highly effective, further reducing the need for a backward pass. As a result, the additional computational cost of the backward scan brings little benefit, resulting in only slight gains in segmentation accuracy.

The Dual-scan (forward + random) method, which introduces a random scan alongside a forward pass, is designed to capture more complex sequential relationships that may not be evident in standard scanning orders. Although randomizing the scanning order can diversify the captured spatial relationships, it also risks compromising the spatial coherence of the data, which could explain why its performance is on par with Dual-scan (forward + backward) rather than exceeding it. This method may identify some complex dependencies but does so at the cost of distorting the structural priors of medical images, which is essential for precise segmentation.

Tri-scan obtains the best results, achieving the highest Dice scores across all datasets. Scanning in three directions (left-right, up-down, front-back) effectively mitigates the spatial discontinuity that can arise from sequential scanning, ensuring a more thorough capture of spatial relationships across the 3D volume. This is particularly beneficial for tasks like TotalSeg, where the segmentation involves 117 classes and complex spatial relationships are needed be captured. The improvement in TotalSeg is more pronounced since the complexity of the task, which requires distinguishing between a wide variety of structures, benefits more from a comprehensive multi-directional scan. Despite this, the trade-off is clear—the higher computational cost makes Tri-scan limitations for resource-constrained applications.

## 7 COMPARISON WITH ADVANCED BASELINES

To further validate the correctness of the aforementioned conclusions, we integrate all the validated strategies into a unified model and compare its performance against advanced baselines. Specifically, we 1) replace the Transformer with Mamba while modifying the 1D depthwise convolution to 3D

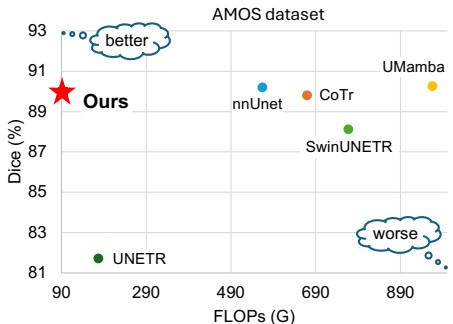 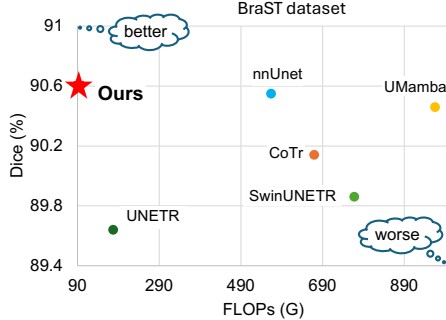

Figure 4: Segmentation Dice scores (higher is better) and FLOPs (lower is better) of Ulike-Mamba_3dMT and against advanced baselines on AMOS and BraTS test sets.

depthwise convolution in the Mamba layer, 2) adopt the multi-scale strategy, *i.e.*, MSv4, and 3) adopt tri-directional scanning, *i.e.*, Tri-scan, to better capture comprehensive spatial relationships in 3D volumetric data. We denote this network as UlikeMamba_3dMT. The specific architecture details can be found in Fig. 6 of Supplementary.

The results in Fig. 4 demonstrate the superiority of UlikeMamba_3dMT over other advanced networks on both AMOS and BraTS datasets. UlikeMamba_3dMT achieves the competitive Dice scores (89.95 in AMOS and 90.60 in BraTS) with the lowest computational cost (93.09G FLOPs), outperforming leading the CNN-based network nnUNet (Isensee et al., 2021), Transformer-based networks such as CoTr (Xie et al., 2021), UNETR (Hatamizadeh et al., 2022), and Swin-UNETR (Hatamizadeh et al., 2021), as well as the existing Mamba-based networks U-Mamba (Ma et al., 2024), which simply integrates Mamba with CNNs. Our UlikeMamba_3dMT integrates Mamba's SSM with 3D depthwise convolutions, the proposed multi-scale modeling, and the designed Tri-scan strategy, proving highly effective by delivering competitive accuracy (Dice scores) while maintaining computational efficiency. This establishes UlikeMamba_3dMT as a new benchmark in 3D medical image segmentation.

## 8 CONCLUSION AND DISCUSSION

In this study, we present a comprehensive exploration of Mamba networks for 3D medical image segmentation, addressing three critical questions: Can Mamba replace Transformers for long-range dependency modeling? Can it improve multi-scale representation learning? Are complex scanning strategies necessary? Our results show that Mamba, with its state space model, not only serves as an effective replacement for Transformers but also offers superior computational efficiency. By modifying the Mamba layer with 3D depthwise convolutions, we address the unique challenges of 3D medical imaging, ensuring better preservation of volumetric spatial coherence and achieving high segmentation accuracy. We further demonstrate the power of Mamba in enhancing multi-scale representation learning by introducing MSv4, a multi-scale modeling strategy that captures both fine-grained details and global context. This capability is particularly important in complex segmentation tasks like those presented in segmenting 117 anatomical structures, where multiple anatomical structures of varying sizes must be accurately delineated. Besides, our study critically evaluates the necessity of complex scanning strategies. While simpler approaches like single-scan generally suffice, the Tri-scan approach significantly improves performance in the most challenging cases by better capturing comprehensive spatial relationships across all dimensions of 3D data.

The UlikeMamba_3dMT network, which integrates all these validated strategies—3D depthwise convolutions, multi-scale modeling, and Tri-scan—establishes a new benchmark for 3D medical image segmentation. It consistently outperforms advanced models such as nnUNet, CoTr, UNETR, SwinUNETR, and U-Mamba, achieving competitive Dice scores with reduced computational complexity. These findings underscore the potential of Mamba-based architectures to push the boundaries of volumetric medical image segmentation, offering both greater accuracy and computational efficiency. Future research should explore further optimizations, *e.g.*, adaptive multi-scan mechanisms, to extend Mamba's applicability across a wider range of medical imaging tasks.

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

# A APPENDIX

## A.1 IMPLEMENTATION DETAILS

In Table 4, we outline the implementation details for the three datasets, covering aspects such as task type, imaging modality, loss function, patch size, batch size, optimizer, learning rate, and maximum iterations. To mitigate overfitting on the training data, we apply online data augmentation techniques, including random rotations, scaling, flipping, the addition of white Gaussian noise, Gaussian blurring, brightness and contrast adjustments, low-resolution simulation, and Gamma transformation (Isensee et al., 2021).

Table 4: Implementation details for three datasets. Dice: Dice loss; CE: Cross-entropy loss.

| Dataset | AMOS | TotalSegmentator | BraTS |
|---|---|---|---|
| Tasks | 15 abdominal organs | 117 anatomical structures | 3 brain tumors |
| Modality | 3D CT | 3D CT | 3D MRI (Four modalities) |
| Loss | Dice+CE | Dice+CE | Dice+CE |
| Patch size | $64 \times 192 \times 160$ | $128 \times 128 \times 128$ | $128 \times 128 \times 128$ |
| Online augmentation | ✓ | ✓ | ✓ |
| Optimizer | AdamW | AdamW | AdamW |
| Learning rate | 0.0001 | 0.0001 | 0.0001 |
| Batch size | 2 | 2 | 2 |
| Max. iterations | 250,000 | 250,000 | 250,000 |

## A.2 ARCHITECTURES OF ULIKEMAMBA_3D AND ULIKETRANS_SRA

Figure 5 shows the detailed configurations of the UlikeMamba_3d and UlikeTrans_SRA networks.

## A.3 OUR PROPOSED MAMBA LAYER IN ULIKEMAMBA_3DMT

Figure 6 shows the details of our proposed Mamba layer in UlikeMamba_3dMT.

**Left table — UlikeMamba_3d**

| Layer_name | | UlikeMamba_3d | Output Size |
|---|---|---|---|
| 3D Conv Downsample | | Conv: K=7, C=32, S=(1,2,2) | $D \times \frac{H}{2} \times \frac{W}{2}$ |
| ES1 | 3D Conv Downsample | Conv: K=3, C=48, S=2 | $\frac{D}{2} \times \frac{H}{4} \times \frac{W}{4}$ |
| | Mamba Layer | Linear1: 48 → 2×48
Linear2: 48 → 2×48
DWConv: K=3×3×3, C=2×48 ×2
SSM, C=2×48
Multiplicative
Linear3: 2×48 → 48 | $\frac{D}{2} \times \frac{H}{4} \times \frac{W}{4}$ |
| ES2 | 3D Conv Downsample | Conv: K=3, C=128, S=2 | $\frac{D}{4} \times \frac{H}{8} \times \frac{W}{8}$ |
| | Mamba Layer | Linear1: 128 → 2×128
Linear2: 128 → 2×128
DWConv: K=3×3×3, C=2×128 ×3
SSM, C=2×128
Multiplicative
Linear3: 2×128 → 128 | $\frac{D}{4} \times \frac{H}{8} \times \frac{W}{8}$ |
| ES3 | 3D Conv Downsample | Conv: K=3, C=256, S=2 | $\frac{D}{8} \times \frac{H}{16} \times \frac{W}{16}$ |
| | Mamba Layer | Linear1: 256 → 2×256
Linear2: 256 → 2×256
DWConv: K=3×3×3, C=2×256 ×4
SSM, C=2×256
Multiplicative
Linear3: 2×256 → 256 | $\frac{D}{8} \times \frac{H}{16} \times \frac{W}{16}$ |
| ES4 | 3D Conv Downsample | Conv: K=3, C=512, S=2 | $\frac{D}{16} \times \frac{H}{32} \times \frac{W}{32}$ |
| | Mamba Layer | Linear1: 512 → 2×512
Linear2: 512 → 2×512
DWConv: K=3×3×3, C=2×512 ×3
SSM, C=2×512
Multiplicative
Linear3: 2×512 → 512 | $\frac{D}{16} \times \frac{H}{32} \times \frac{W}{32}$ |
| DS1 | 3D Conv Upsample | TransposeConv: K=2, C=256, S=2 | $\frac{D}{8} \times \frac{H}{16} \times \frac{W}{16}$ |
| | Mamba Layer | Linear1: 256 → 2×256
Linear2: 256 → 2×256
DWConv: K=3×3×3, C=2×256 ×3
SSM, C=2×256
Multiplicative
Linear3: 2×256 → 256 | $\frac{D}{8} \times \frac{H}{16} \times \frac{W}{16}$ |
| DS2 | 3D Conv Upsample | TransposeConv: K=2, C=128, S=2 | $\frac{D}{4} \times \frac{H}{8} \times \frac{W}{8}$ |
| | Mamba Layer | Linear1: 128 → 2×128
Linear2: 128 → 2×128
DWConv: K=3×3×3, C=2×128 ×4
SSM, C=2×128
Multiplicative
Linear3: 2×128 → 128 | $\frac{D}{4} \times \frac{H}{8} \times \frac{W}{8}$ |
| DS3 | 3D Conv Upsample | TransposeConv: K=2, C=48, S=2 | $\frac{D}{2} \times \frac{H}{4} \times \frac{W}{4}$ |
| | Mamba Layer | Linear1: 48 → 2×48
Linear2: 48 → 2×48
DWConv: K=3×3×3, C=2×48 ×3
SSM, C=2×48
Multiplicative
Linear3: 2×48 → 48 | $\frac{D}{2} \times \frac{H}{4} \times \frac{W}{4}$ |
| 3D Conv Upsample | | TransposeConv: K=2, C=32, S=2 | $D \times \frac{H}{2} \times \frac{W}{2}$ |
| Segmentation Head | | Conv: K=3, C=32, S=1
Upsample: S=(1,2,2)
Conv: K=1, C=Num_classes, S=1 | $D \times H \times W$ |

**Right table — UlikeTrans_SRA**

| Layer_name | | UlikeTrans_SRA | Output Size |
|---|---|---|---|
| 3D Conv Downsample | | Conv: K=7, C=32, S=(1,2,2) | $D \times \frac{H}{2} \times \frac{W}{2}$ |
| ES1 | 3D Conv Downsample | Conv: K=3, C=48, S=2 | $\frac{D}{2} \times \frac{H}{4} \times \frac{W}{4}$ |
| | Transformer Layer | R = 6
H = 1  ×2
E = 4 | $\frac{D}{2} \times \frac{H}{4} \times \frac{W}{4}$ |
| ES2 | 3D Conv Downsample | Conv: K=3, C=128, S=2 | $\frac{D}{4} \times \frac{H}{8} \times \frac{W}{8}$ |
| | Transformer Layer | R = 4
H = 2  ×3
E = 4 | $\frac{D}{4} \times \frac{H}{8} \times \frac{W}{8}$ |
| ES3 | 3D Conv Downsample | Conv: K=3, C=256, S=2 | $\frac{D}{8} \times \frac{H}{16} \times \frac{W}{16}$ |
| | Transformer Layer | R = 2
H = 4  ×4
E = 4 | $\frac{D}{8} \times \frac{H}{16} \times \frac{W}{16}$ |
| ES4 | 3D Conv Downsample | Conv: K=3, C=512, S=2 | $\frac{D}{16} \times \frac{H}{32} \times \frac{W}{32}$ |
| | Transformer Layer | R = 1
H = 8  ×3
E = 4 | $\frac{D}{16} \times \frac{H}{32} \times \frac{W}{32}$ |
| DS1 | 3D Conv Upsample | TransposeConv: K=2, C=256, S=2 | $\frac{D}{8} \times \frac{H}{16} \times \frac{W}{16}$ |
| | Transformer Layer | R = 2
H = 8  ×3
E = 4 | $\frac{D}{8} \times \frac{H}{16} \times \frac{W}{16}$ |
| DS2 | 3D Conv Upsample | TransposeConv: K=2, C=128, S=2 | $\frac{D}{4} \times \frac{H}{8} \times \frac{W}{8}$ |
| | Transformer Layer | R = 4
H = 4  ×4
E = 4 | $\frac{D}{4} \times \frac{H}{8} \times \frac{W}{8}$ |
| DS3 | 3D Conv Upsample | TransposeConv: K=2, C=48, S=2 | $\frac{D}{2} \times \frac{H}{4} \times \frac{W}{4}$ |
| | Transformer Layer | R = 6
H = 2  ×3
E = 4 | $\frac{D}{2} \times \frac{H}{4} \times \frac{W}{4}$ |
| 3D Conv Upsample | | TransposeConv: K=2, C=32, S=2 | $D \times \frac{H}{2} \times \frac{W}{2}$ |
| Segmentation Head | | Conv: K=3, C=32, S=1
Upsample: S=(1,2,2)
Conv: K=1, C=Num_classes, S=1 | $D \times H \times W$ |

Figure 5: **Left**: detailed configurations of UlikeMamba_3d network. Here, 'K': kernel size of Conv, DWConv or TransposeConv; 'C': number of channels; and 'S': stride. **Right**: Detailed configurations of UlikeTrans_SRA network. Here, 'R': reduction ratio of SRA; 'H': head number of SRA; and 'E': expansion ratio of FFN.

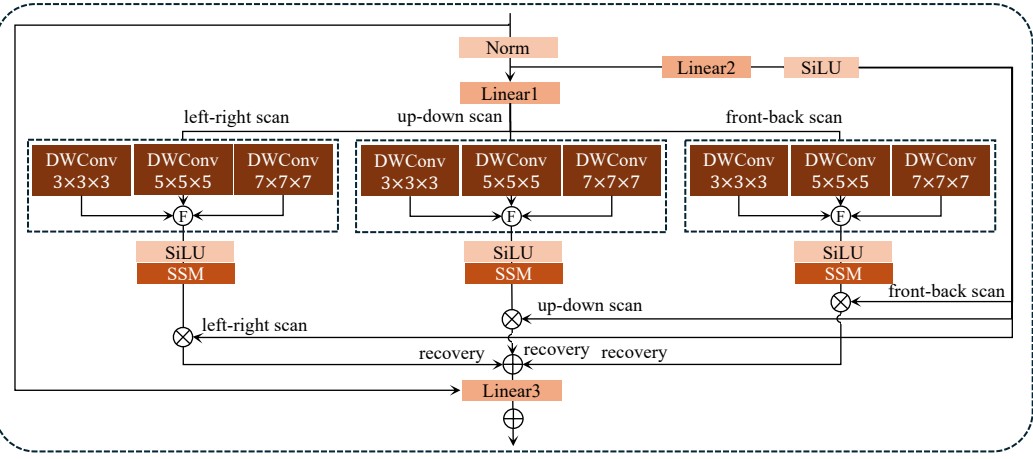

Figure 6: Our proposed Mamba layer in UlikeMamba_3dMT, which modifies the original 1D depthwise convolution to 3D depthwise convolution, embraces a multi-scale strategy, and incorporates tri-directional scanning to more effectively capture comprehensive spatial relationships in 3D volumetric data.

