# OpenReview forum: "What can Mamba do for 3D Volumetric Medical Image Segmentation?"
_ICLR.cc/2025/Conference — Submitted to ICLR 2025_

### Official Review · Reviewer_Njbp · 2024-11-01

**Soundness:** 2
**Presentation:** 3
**Contribution:** 2
**Rating:** 3
**Confidence:** 4

**Summary:**

The paper presents a Mamba-based U-shape model for 3D volumetric medical image segmentation. By comparison with TransformerSRA-based U-shape models, Mamba-based model can achieve a better performance. The authors replace the original 1D depthwise convolution by multiple multi-scale depthwise convolutions. Meanwhile, the authors present the Tri-scan outperforms the exiting scan methods for SSM. This paper is easy to follow.

**Strengths:**

1. The authors compare Mamba-based U-shape model with a simpler Transformer-based model.
2. The authors design a multi-scale depthwise convolution before SSM.
3. The authors present a Tri-scan design for 3D medical images.
4. This paper is easy to follow.

**Weaknesses:**

1. To demonstrate the capabilities of modeling long-term dependencies, the authors develop a baseline Transformer-based model utilizing a simpler spatial-reduction attention (SRA) layer. While the vanilla Transformer causes the out-of-memory issue (Please refer to [1] for details how it addresses memory issues.), it does not convincingly demonstrate whether Mamba can achieve superior performance.
2. Even if the performance of Mamba-based models outperform Transformer-based models in metrics, it remains unclear whether Mamba effectively models long-term dependencies. The authors could strengthen their claim by presenting activation maps or receptive fields. In VMamba [3], Figure 5 provides a feasible activation map for individual scanning path of Mamba. The reason may be a strong inductive bias on medical image segmentation tasks. [2] indicates Transformer-based models can not achieve better performance comparied with CNN-based models.
3. Although the authors replace the original 1D DWConv with a multi-scale 3D DWConv and design a Tri-scan, the authors do not clarify why we need Mamba for medical images. Is it necessary to use Mamba for medical images. The authors could provide how SSM helps modeling 3D medical images, such as activation maps.
4. For AMOS22,  it is advisable to evaluate your models using the testing dataset on the official challenge website. Can the authors provide the results for the testing dataset?
5. Lack of comparisons with existing Mamba-based methods, such as UMamba, SegMamba, and VM-UNet.
6. While the author provide Figure 4 about the comparison with existing SOTA methods, it is better if the author can provide a quantitative table to compare with exsiting SOTA methods. Meanwhile, can the authors provide more recent methods, such as nnUNet_ResEnc [2], nnFormer [8], 3D UX-Net [7] and MedNeXt [9].


[1] Zhou, Hong-Yu, et al. "nnformer: Interleaved transformer for volumetric segmentation." arXiv preprint arXiv:2109.03201 (2021).

[2] Isensee, Fabian, et al. "nnu-net revisited: A call for rigorous validation in 3d medical image segmentation." International Conference on Medical Image Computing and Computer-Assisted Intervention. Cham: Springer Nature Switzerland, 2024.

[3] Liu, Yue, et al. "Vmamba: Visual state space model." arXiv preprint arXiv:2401.10166 (2024)

[4] Ma, Jun, Feifei Li, and Bo Wang. "U-mamba: Enhancing long-range dependency for biomedical image segmentation." arXiv preprint arXiv:2401.04722 (2024).

[5] Xing, Zhaohu, et al. "Segmamba: Long-range sequential modeling mamba for 3d medical image segmentation." International Conference on Medical Image Computing and Computer-Assisted Intervention. Cham: Springer Nature Switzerland, 2024.

[6] Ruan, Jiacheng, and Suncheng Xiang. "Vm-unet: Vision mamba unet for medical image segmentation." arXiv preprint arXiv:2402.02491 (2024).

[7] Lee, Ho Hin, et al. "3d ux-net: A large kernel volumetric convnet modernizing hierarchical transformer for medical image segmentation." arXiv preprint arXiv:2209.15076 (2022).

[8] Zhou, Hong-Yu, et al. "nnformer: Interleaved transformer for volumetric segmentation." arXiv preprint arXiv:2109.03201 (2021).

[9] Roy, Saikat, et al. "Mednext: transformer-driven scaling of convnets for medical image segmentation." International Conference on Medical Image Computing and Computer-Assisted Intervention. Cham: Springer Nature Switzerland, 2023.

**Questions:**

see the weakness

---

### Official Review · Reviewer_3gVd · 2024-11-03

**Soundness:** 2
**Presentation:** 2
**Contribution:** 2
**Rating:** 3
**Confidence:** 5

**Summary:**

This work analyzes Mamba-based 3D medical image segmentation networks and proposes task-specific modifications with depth-wise convolutions et al. Experiments on three 3D datasets demonstrate that the new networks provide better efficiency with secarifying the segmentation accuracy.

**Strengths:**

- Detailed comparisons between Mamba-based and Transformer-based 3D segmentation networks on three 3D medical image segmentation datasets
- Improve the computation efficiency of existing UlikeMamba substantially

**Weaknesses:**

- Statistical analysis is missing. In general, ~1% improvements in DSC have no significant difference. Please report p-value under Wilcoxon signed rank test.
- Boundary-related metrics are missing. Please follow the consensus to select the metrics: https://www.nature.com/articles/s41592-023-02151-z For example, Normalized surface distance can be a great candidate. Moreover, please discuss how these metrics might provide insights into the model's performance that Dice scores alone do not capture.
- Lack of latest baseline models. The implementation was based on nnU-Net framework. However, the recommended baseline models have been changed to ResU-Net by the author team.

**Questions:**

- The employed nnUNet baseline is outdated. How does UlikeMamba 3dMT compare to nnUNet-ResM/L/XL? (https://link.springer.com/chapter/10.1007/978-3-031-72114-4_47)
- Are the claimed improvements statistically significant under Wilcoxon signed rank test? Furthermore, please discuss the practical significance of their improvements in the context of clinical applications.
- TotalSegmentation divides the 117 organs into multiple groups. Could you please separately report the performance for each organ group? It would be great to discuss how the performance varies across organ groups and what insights this might provide about the strengths and limitations of the approach. Additionally, could you please analyze whether certain organ groups benefit more from the Mamba-based architecture compared to others, and discuss potential reasons for any observed differences?

---

### Official Review · Reviewer_e7hy · 2024-11-03

**Soundness:** 2
**Presentation:** 2
**Contribution:** 2
**Rating:** 3
**Confidence:** 4

**Summary:**

The author introduces UlikeMamba_3dMT, a Mamba-based model for 3D medical image segmentation tasks, comparing it with UlikeTransformer, a transformer-based counterpart. UlikeMamba_3dMT incorporates multi-scale depthwise convolutions and a Tri-scan strategy to capture spatial dependencies in 3D data. Through evaluations on AMOS, TotalSegmentator, and BraTS datasets, the authors claim computational efficiency and competitive segmentation accuracy for UlikeMamba_3dMT as a substitute for UlikeTransformer models.

**Strengths:**

1. This paper provides a direct comparison between UlikeMamba and its transformer-based counterpart, UlikeTransformer. This head-to-head evaluation highlights both the strengths and limitations of UlikeMamba.
2. Mamba’s linear complexity with sequence length potentially offers a promising approach for processing high-resolution 3D images without high computational resources.

**Weaknesses:**

1. The overall novelty of the proposed method is limited. Key components of UlikeMamba, such as multi-scale depthwise convolutions and multi-way scanning, lack originality. The multi-scale depthwise convolutions was previously introduced in EMCAD [1], while multi-way scanning strategies have been explored in VisionMamba [2] and VMamba [3]. While the Tri-scan offers only incremental improvements, the paper does not clearly differentiate how UlikeMamba's method significantly advances these existing strategies. The potential disadvantages of large-kernel (e.g., 3x3 or 5x5, 7x7) depthwise convolutions, such as the loss of inter-channel relationships, are also not addressed.

[1] Rahman, M.M., Munir, M. and Marculescu, R., 2024. Emcad: Efficient multi-scale convolutional attention decoding for medical image segmentation. In Proceedings of the IEEE/CVF Conference on Computer Vision and Pattern Recognition (pp. 11769-11779).
[2] Zhu, L., Liao, B., Zhang, Q., Wang, X., Liu, W. and Wang, X., 2024. Vision mamba: Efficient visual representation learning with bidirectional state space model. arXiv preprint arXiv:2401.09417.
[3] Liu, Y., Tian, Y., Zhao, Y., Yu, H., Xie, L., Wang, Y., Ye, Q. and Liu, Y., 2024. VMamba: Visual State Space Model. arXiv preprint arXiv:2401.10166.

2. Essential baseline comparisons are missing, especially with state-of-the-art models like 3D UX-NET [4], which uses large-kernel depthwise convolutions, and other Mamba-based models such as SegMamba [5] and SwinUMamba [6]. The incorporation of these missing methods in comparison will strengthen UlikeMamba_3dMT's performance in a broader context. Additionally, final evaluations completely missing on TotalSegmentor dataset with UlikeMamba_3dMT, the incorporation of which will strengthen the contribution of the paper.

[4] Lee, H.H., Bao, S., Huo, Y. and Landman, B.A., 2022. 3d ux-net: A large kernel volumetric convnet modernizing hierarchical transformer for medical image segmentation. arXiv preprint arXiv:2209.15076.

[5] Xing, Z., Ye, T., Yang, Y., Liu, G. and Zhu, L., 2024, October. Segmamba: Long-range sequential modeling mamba for 3d medical image segmentation. In International Conference on Medical Image Computing and Computer-Assisted Intervention (pp. 578-588). Cham: Springer Nature Switzerland.

[6] Liu, J., Yang, H., Zhou, H.Y., Xi, Y., Yu, L., Li, C., Liang, Y., Shi, G., Yu, Y., Zhang, S. and Zheng, H., 2024, October. Swin-umamba: Mamba-based unet with imagenet-based pretraining. In International Conference on Medical Image Computing and Computer-Assisted Intervention (pp. 615-625). Cham: Springer Nature Switzerland.

3. The paper lacks a theoretical explanation for Mamba’s effectiveness over transformers in 3D segmentation, thus relying solely on experimental results for validation. Without an analytical foundation or insight into why Mamba performs well in 3D segmentation, this work feels similar to existing Mamba-based methods [5,6] that only present experimental validation.

4. Important computational metrics, such as memory usage and latency, are missing from the analysis. Furthermore, the paper is hard to follow due to its organization, with experimental results scattered throughout the methodology. A clearer structure and a table summarizing parameters, FLOPs, latency, and dataset performance would improve readability and transparency.

5. The Tri-scan strategy results in only little improvements in Dice scores, while significantly increasing computational costs. The minimal accuracy gains may not justify the added complexity, particularly without evidence that demonstrates the necessity of these trade-offs in practical scenarios.

6. The multi-scale convolutions contributes minimally to UlikeMamba’s performance, especially on smaller tasks (e.g., BraTS, where MSv4 shows reduced performance). The authors should consider a few other 3D binary segmentation datasets to validate that this is not a common limitation of UlikeMamba in case of smaller tasks.

7. The authors claim that UlikeMamba captures long-range dependencies effectively, yet there is no evidence supporting this beyond the reported segmentation accuracy. This improvement in accuracy may not directly result from enhanced long-range dependency modeling, and further clarification on this aspect would strengthen the paper.

**Questions:**

1. How would UlikeMamba perform if other commonly used transformer blocks, such as the SwinTransformer block, were used instead of SRA attention? Testing with a popular transformer design could clarify the suitability of Mamba as an alternative.

2. Can the authors provide visual or qualitative results to validate the claim that UlikeMamba effectively captures fine-grained details and larger anatomical structures? A comparison of multi-scale and single-scale configurations could justify the use of the multi-scale approach.

3. Why do author use tri-scan strategy, why not other tri-scan (e.g., right-left, down-up, and back-front) or hexa-scan strategy (e.g., left-right, right-left, up-down, down-up, front-back, and back-front)? A comparative study of these alternatives could help justify the use of Tri-scan in the model.

---

### Official Review · Reviewer_UrTu · 2024-11-04

**Soundness:** 3
**Presentation:** 3
**Contribution:** 2
**Rating:** 5
**Confidence:** 3

**Summary:**

This paper presents a comprehensive investigation into Mamba’s capabilities in 3D medical image segmentation, especially the comparison with Transformer-based models. Specifically, three interesting issues are detailedly discussed with extensive experiments. 1) Mamba’s ability to replace Transformers. 2) Mamba’s capacity to enhance multi-scale representation learning. 3) Whether complex multi-way scanning strategies are necessary?

**Strengths:**

1)  Using AMOS, TotalSegmentator, and BraTS datasets for evaluation, authors claim that UlikeMamba outperforms Ulike-Trans in both accuracy and efficiency.
2)  Simpler scanning methods often suffice, with Tri-scan proving advantageous in more complex scenarios.
3)  The Mamba-based segmentation model in this work achieves high performance compared with nnUNet with less computational costs.

**Weaknesses:**

1) There is a missing literature review of Transformer-based models in the part of related works.
2) Authors mainly compare the network structure of Mamba-based compared with Transformer-based models. However, according to a recent study by [1], nnUNet/MedNeXt outperform almost all the Transformer-based and Mamba-based models. Thus, I am a little bit confused about the quantitative results in this paper.
3) nnUNet is a strong backbone for a large variety of medical datasets, and Transformer-based models fail to outperform nnUNet in almost all MICCAI Challenges. I am wondering if Mamba-based models could do that, especially on recent challenging datasets, such as Topcow24, Aortaseg24?

[1] nnU-Net Revisited: A Call for Rigorous Validation in 3D Medical Image Segmentation, MICCAI 2024.

**Questions:**

Please see the weakness part.

---

### Meta-Review · Area_Chair_6PwA · 2024-12-20

**Metareview:**

Major Concerns:
1. Limited Novelty. Key components (multi-scale convolutions, multi-way scanning) are borrowed from existing work.
2. Evaluation Issues:
   - Missing critical baseline comparisons (3D UX-NET, SegMamba, SwinUMamba)
   - Lacks statistical analysis (no p-values for performance differences)
   - No evidence supporting claims about long-range dependency modeling
   - Confusion about performance claims given recent studies showing nnUNet/MedNeXt superiority

**Additional Comments On Reviewer Discussion:**

The reviewers suggest that while UlikeMamba shows some promise, the paper needs more rigorous validation, clearer theoretical justification, and more comprehensive comparisons with state-of-the-art methods to substantiate its claims.

---

### Decision · Program_Chairs · 2025-01-22

Reject